# Reduced Sodium in White Brined Cheese Production: Artificial Neural Network Modeling for the Prediction of Specific Properties of Brine and Cheese during Storage

Katarina Lisak Jakopović [ID], Irena Barukčić Jurina *[ID], Nives Marušić Radovčić [ID], Rajka Božanić [ID] and Ana Jurinjak Tušek

Department of Food Engineering, Faculty of Food Technology and Biotechnology, University of Zagreb, Pierottijeva 6, 10000 Zagreb, Croatia; klisak@pbf.unizg.hr (K.L.J.); nmarusic@pbf.hr (N.M.R.); rbozan@pbf.hr (R.B.); ana.tusek.jurinjak@pbf.unizg.hr (A.J.T.)
* Correspondence: irena.barukcic@pbf.unizg.hr; Tel.: +385-1-4605-017

**Abstract:** Background: White brined cheese is one of the most frequently consumed cheeses that is accepted among a large group of consumers, which is largely related to its unique sensory properties, which are characterized by specific technological processes including ripening in the brine. Thus, white brined cheese contains a high amount of NaCl, and frequent consumption might lead to excessive sodium intake, which nowadays, presents a global problem. Consequently, food industries have developed reduced sodium products by substitutional salts. Furthermore, various studies have indicated that increased sodium intake via the diet can be associated with cardiovascular diseases, a risk of digestive system cancer, obesity, and other conditions. Calcium salts (citrate and lactate) are safe for human health and can be added to various foods according to the *quantum satis* rule. The present study aimed to partially replace NaCl with Ca-citrate and Ca-lactate in proportions of 25% and 50%. Additionally, the study presents the possibility of applying Artificial Neural Network (ANN) models for the prediction of some brine and cheese properties. Methods: White brined cheese with substitutional salts in brine (25% and 50% Ca-citrate and 25% and 50% Ca-lactate) were produced and compared to the control cheese ripened in conventionally applied NaCl brine. The acidity, total dissolved solids, salt amount, conductivity, color, and textural and sensory properties were determined over the 28 days of cold storage. Results: The substitution of NaCl with Ca-citrate and Ca-lactate is promising for sodium reduction in white brined cheese, whereby the physical and chemical properties remain acceptable. The best sensory score gain occurred with a substitution of 25% NaCl with Ca-citrate. Furthermore, ANN models can be employed to predict brine and cheese properties during storage.

**Keywords:** white brined cheese; substitutional salts; Ca-citrate; Ca-lactate; artificial neural network modeling

## 1. Introduction

Excessive sodium intake is most often associated to hypertension [1], osteoporosis [2–4], and the incidence of kidney stones [3], as well as chronic kidney disease [5]. Over the past decade, consumers' concern about sodium in processed foods has increased, and the average consumer has become more aware of its adverse health effects. The World Health Organization (WHO) recommends a daily salt intake (sodium chloride) of below 5 g for adults, which is equivalent to 2 g of sodium. Most people consume higher amounts of sodium chloride, an average of 9 to 15 g per day, which is three times higher than recommended. Concerned about this situation, WHO member states adopted strategies to reduce the salt intake of their populations by 30% by 2025 [6]. It is estimated that 75% of sodium intake comes from processed foods [7].

Feta-type cheeses and similar cheeses like white brined cheese are white cheeses with a characteristic sour and salty taste due to the high NaCl concentration in the brine. Additionally, these cheeses have a homogeneous cross-section and smooth texture with no or a few cracks that can occur as a consequence of the mechanical treatment of the curd. Feta cheese originates from Greece and is traditionally produced from sheep milk [8]. White brined cheese may be manufactured from any type of milk with different fat contents. The cheese ripens in the brine with 8–10% salt, which usually takes 15–25 days, depending on the temperature at which it is stored [9–13]. Cheese can be stored in brine for up to one year [8]. White brined cheese can contain up to 7% NaCl; thus, reducing salt is necessary for improving public health outcomes.

However, salt has many roles in cheese production, such as the modulation of the physicochemical and biochemical properties of cheese during ripening, maintenance of low water activity, which controls the development of micro-organisms, and, most importantly, the conditioning of the sensory characteristics of the final products. For all of the mentioned reasons, it is difficult to achieve optimal salt reduction [7]. Additionally, salt can affect the rheological (elasticity, breaking point, adhesion, hardness) as well as physical properties of cheese, such as its ability to be crushed, ground, and cut [10,13].

According to the literature, there are two methods for lowering the sodium amount. The first implies the reduction of salt in processed foods, which can lead to microbiological spoilage. The second method includes the total or partial replacement of NaCl by other salts. The replacement of sodium chloride with spices and other salts, such as potassium chloride, calcium chloride, magnesium chloride, potassium lactate, and potassium phosphate, is widespread in the food industry. Numerous studies that focused on potassium chloride (KCl) usage resulted in a significant reduction in sodium without negative effects on the cheese quality [14–17]. Thereby, salt replacement in hard, semi-hard, and soft cheeses like Cheddar and mozzarella have most often been researched. Some authors have reported that the substitution of NaCl by KCl increased the bitterness of Cheddar and interrupted lipolysis and proteolysis as well [18]. Berhet [19] reported that KCl-substituted products should be consumed with caution by consumers with renal disease. The substitution of NaCl by $MgCl_2$ or $CaCl_2$ has been reported to be associated with the presence of flavors like sour, bitter, metallic, and soapy and thus decreases the acceptability of Cheddar [20,21]. Gore et al. [22] reported a new method for decreasing Na by partly substituting NaCl with alkalinizing organic Ca salts. They produced a blue-veined cheese with 75% of NaCl substituted by calcium lactate to obtain cheese with approximately 33% less Na than the control cheese that was salted only with fine NaCl. Besides that, substitution with calcium citrate increased the citrate content by 410% compared with the control cheese. There were no significant differences in bitter or salty perceptions between the produced cheeses compared with the control cheese. Apart from the study conducted by Gore et al. [22], so far, calcium lactate has been used for calcium fortification of milk (39%), cottage cheese, and yoghurt (36%) [23,24]. Therefore, this substitute could be of both technological and nutritional interest. To our knowledge, there is little research related to the sodium chloride substitution in feta-type cheeses. Katsiari et al. [25] applied feta-type cheeses produced with a highly acceptable quality and without significant differences from the control cheese. Thereby, they used brines with a 3:1 or 1:1 (*w/w*) mixture of NaCl and KCl instead of using NaCl alone, and these contained about 25 and 50% less sodium, respectively.

Considering the importance of lower-sodium food production and the lack of research on low-sodium white brined cheese, the objective of this research was to investigate the possibility of partially replacing sodium chloride in brine with calcium citrate and calcium lactate and to determine the effects on physicochemical and sensory parameters during white brined cheese ripening for 28 days. Artificial Neural Network modeling was used for the prediction of specific properties of brine and cheese during the storage period.

## 2. Materials and Methods

### 2.1. Materials

Raw cow's milk for white brined cheese production was obtained from a local dairy farm. During the production of white brined cheese, the mesophilic starter culture CHN-22 (DuPont, Paris, France) was used. Rennet (enzyme preparation SIRIS, Lub d.o.o., Split, Croatia) was added according to the manufacturer's instructions. The used additives were $CaCl_2$ and $KNO_3$, 9% acetic acid for lowering the pH of the milk, and NaCl salt as well as the substitute salts Ca-citrate and Ca-lactate.

### 2.2. White Brined Cheese Production

Each batch of cheese was produced from 12 L of raw milk. After sampling, the milk was standardized by skimming in the separator (Libra tehničar d.o.o., Zareb, Croatia) to about 3% milk fat and pasteurized at 72 °C/15 s. Subsequently, the pasteurized milk was cooled to 34–36 °C (optimum temperature for the incubation of the starter culture), and $CaCl_2$ (10–20 g/100 L of milk), $KNO_3$ (0.01%), and the starter culture (0.5–1%) were added. Incubation was performed at 34–36 °C for 45 min. After that, 9% acetic acid was added to lower the pH value to about 6.0 pH units. In the prepared matured milk, rennet was added and put into an incubator for 30 min at 35 °C. After that, the curd was cut into 1–2 cm cubes and gently stirred for better whey extraction. Curd was transferred to the strainer with cheese coats for 60 min for draining, followed by draining under its weight for another 60 min. Curd was transferred to moulds and pressed with 5 kg weights (then, for several hours with 10 kg weights), and the total process took place at 16 °C for 18–24 h with several rotations of the cheese in the mould. The shaped cheese was removed from the mould, cut into 1.5 cm thick slices (portioning at about 100 g of cheese), and placed in glass containers with brine. The cheese in the brine ripened and was cold-stored for 28 days, and analyses were performed every seven days (before putting the cheese into brine (0) and on the 7th, 14th, 21st, and 28th days).

### 2.3. Brine Production and Substitutional Salts

All types of brine were prepared as 10% salt solutions (*w/v*), and the pH value was adjusted to 4.7 pH units with 9% acetic acid. The control brine contained 10% NaCl (*w/v*). Substitutional salts were Ca-citrate and Ca-lactate, and their ratios were prepared according to Table 1. After brine preparation, they were pasteurized at 72 °C/15 s and cooled down to the storage temperature. When cheese was immersed, the brine ratio between brine and cheese was 4:1.

**Table 1.** Control (NaCl) and substitutional (Ca-citrate, Ca-lactate) salt ratios (%) in brine.

| Sample | NaCl | Ca-Citrate | Ca-Lactate |
|--------|------|------------|------------|
| BC | 100 | 0 | 0 |
| BCC1 | 75 | 25 | 0 |
| BCC2 | 50 | 50 | 0 |
| BCL1 | 75 | 0 | 25 |
| BCL2 | 50 | 0 | 50 |

BC—control brine with 100% NaCl; BCC1—brine with 75% of NaCl and 25% Ca-citrate; BCC2—brine with 50% of NaCl and 50% Ca-citrate; BCL1—brine with 75% of NaCl and 25% Ca-lactate; BCL2—brine with 50% of NaCl and 50% Ca-lactate.

### 2.4. Brine and Cheese Analyses

White brined cheese was produced in three batches with a delay of 14 days. Cheese and brine analyses were performed on the production day (0 day) and on every th day during the 28 days of the cold storage period. Acidity (pH and titratable acidity by Soxhlet Henkel (°SH)) was determined as described by Lisak Jakopović et al. [26]. Conductivity and total dissolved solids (TDS) were determined by a conductivity meter (TDS/Conductivity/°C

meter, RS 232 CON 200 series, Oakton, Singapore) in accordance with Lisak Jakopović et al. [26]. The salt content was determined following the Mohr method (ISO 1738, 2004) in which chlorides are titrated with silver nitrate solution in the presence of chromate anions. The titration end point is signalled by the appearance of red silver chromate [27].

### 2.5. Cheese Texture Analyses

The texture profile analysis (TPA) of the produced white brined cheese samples was carried out using a texture analyzer (Ametek Lloyd Instruments Ltd., West Sussex, UK). The texture analyzer was equipped with a 50 kg load cell supported by NexygenPlus software. The analysis was performed at room temperature. Prior to the analysis, the samples were cut into cubes, $10 \times 10 \times 10$ mm in size, and conditioned for 2 h at 20 °C. The samples were compressed twice at a crosshead speed of 1 mm/s to 50% deformation (the rest period between cycles was 5 s), and the following parameters were determined from the force–distance curves: hardness (N), adhesive force (N), adhesiveness (Nmm), cohesiveness (N/m), gumminess (N), postponed elasticity (mm), chewiness (Nmm), resistance, breakage (N), and fibrousness (mm) [28].

### 2.6. Color Measurements

CIE surface color parameters (L*: lightness, a*: redness, b*: yellowness) (CIE, 1976) were measured using a Minolta CM-700d spectrophotometer (Minolta, Osaka, Japan) with an illuminant D 65 10° standard observer, 8 mm aperture, and an open cone. Each cheese and brine sample was analyzed in triplicate [28].

### 2.7. Sensory Analyses

A sensory evaluation of white brined cheese was performed by a group of five specially trained panellists using a scoring system of weighted factors on a 20-point scale [29,30]. White brined cheese was cool-stored at 4 °C from the point of production and entry into brine until the point of sampling and evaluation. In a room designed according to the ISO standard 8589:2007 [31] cheese samples were removed from brine, encoded, divided into equal portions, and presented simultaneously to each of the five panellists. Samples were evaluated for their overall appearance, color, consistency, cut, odor, and taste, whereby each attribute could have been rated with scores from 1 to 5. The average score of every attribute was multiplied by a predetermined weighting factor, resulting in a score for each attribute, as follows: overall appearance—2, color—1, consistency—2, cut—3, odor—2 and taste—10. By summarizing the scores of each attribute, a final score for the particular sample was obtained. The maximum score that one sample could obtain was 20 [26].

### 2.8. Statistical Analysis and Modeling

White brined cheese production was repeated three times, and the obtained results of all measurements were expressed as the mean ± standard deviation (SD).

#### 2.8.1. Basic Statistical Analysis

Basic statistical analysis including average values, standard deviations and the Spearman correlations which was performed using Statistica v.10.0 software (StatSoft, Tulsa, OK, USA).

#### 2.8.2. Analysis of Variance

The means of the results were evaluated using an analysis of variance (ANOVA), and Tukey's test was used to compare significant differences ($p < 0.05$) between the physicochemical, textural, and sensory evaluations. The ANOVA was performed using Statistica v.10.0 software (StatSoft, Tulsa, OK, USA).

### 2.8.3. Principle Component Analysis (PCA)

The principle component analysis was performed to identify the effect of the process variable on all analyzed properties of brines and cheeses. In case of the brine properties, the sodium chloride concentration, Ca-citrate concentration, Ca-lactate concentration, and the day of storage were used as the supplementary variables, and the pH, conductivity, TDS, and color coordinates (L*, a*, b*) were used as the variables for the analysis. Furthermore, in case of the cheese properties, the sodium chloride concentration, Ca-citrate concentration, Ca-lactate concentration, day of storage, brine pH, brine conductivity, and brine TDS were supplementary variables, and the cheese pH, °SH, cheese color coordinates (L*, a*, b*), cheese textural properties (hardness, adhesive force, adhesiveness, cohesiveness, gumminess, postponed elasticity, chewiness, resistance, breakage, and fibrousness) and cheese sensory properties (appearance, color, consistency, cut, odor, taste and total) were used as the variables for analysis.

### 2.8.4. Artificial Neural Network (ANN) Modeling

Artificial Neural Network (ANN) modeling was used to predict the properties of brine and the physical and textural properties of cheese and to conduct a sensory evaluation of the produced cheeses over the 28-day storage period.

Multiple Layer Perceptron (MLP) networks were developed using Statistica 14.0 software. (TIBCO®® Statistica, Palo Alto, CA, USA). ANNs consist of three layers: the input, hidden, and output. Four different ANN models were developed:

(i) For the prediction of brine properties (pH, conductivity, TDS, and color coordinates) based on theNaCl concentration, calcium citrate concentration, calcium lactate concentration, and day of storage. The data set for the construction of the ANNs was $75 \times 10$, with seventy-five rows representing the brine samples, four columns representing the model inputs, and six columns representing the model output;

(ii) For the prediction of the physical properties of cheese (pH, °SH, L, and color coordinates) based on the NaCl concentration, calcium citrate concentration, calcium lactate concentration, day of storage, brine pH, and brine conductivity. The data set used for the construction of the ANNs was $75 \times 12$, with seventy-five rows representing the brine samples, seven columns representing the model inputs, and five columns representing the model outputs;

(iii) For the prediction of the textural properties of cheese (hardness, gumminess, chewiness, and breakage) based on the NaCl concentration, calcium citrate concentration, calcium lactate concentration, day of storage, brine pH, and brine conductivity. The data set used for the construction of the ANNs was $75 \times 11$, with seventy-five rows representing the brine samples, seven columns representing the model inputs, and four columns representing the model outputs;

(iv) For the total sensory evaluation based on the NaCl concentration, calcium citrate concentration, calcium lactate concentration, day of storage, brine pH, and brine conductivity. The data set used for the construction of the ANNs was $75 \times 8$, with seventy-five rows representing the brine samples, seven columns representing the model inputs, and one column representing the model outputs

To ensure the development of the relabeled models, cross validation was implemented in Statistica 14.0. (TIBCO®® Statistica, Palo Alto, CA, USA). A subsampling strategy was applied. The back error propagation algorithm was used for model training, and the error function was the sum of squares. The $R^2$ and Root-Mean-Square Error (RMSE) values for training, testing, and validation were used to estimate the performance of the developed models. In order to develop the ANN, data were randomly divided into three categories: network training (70%—53 data points), model test (15%—11 data points), and model validation (15%—11 data points).

The architecture of the artificial network used for prediction is presented in Figure 1.

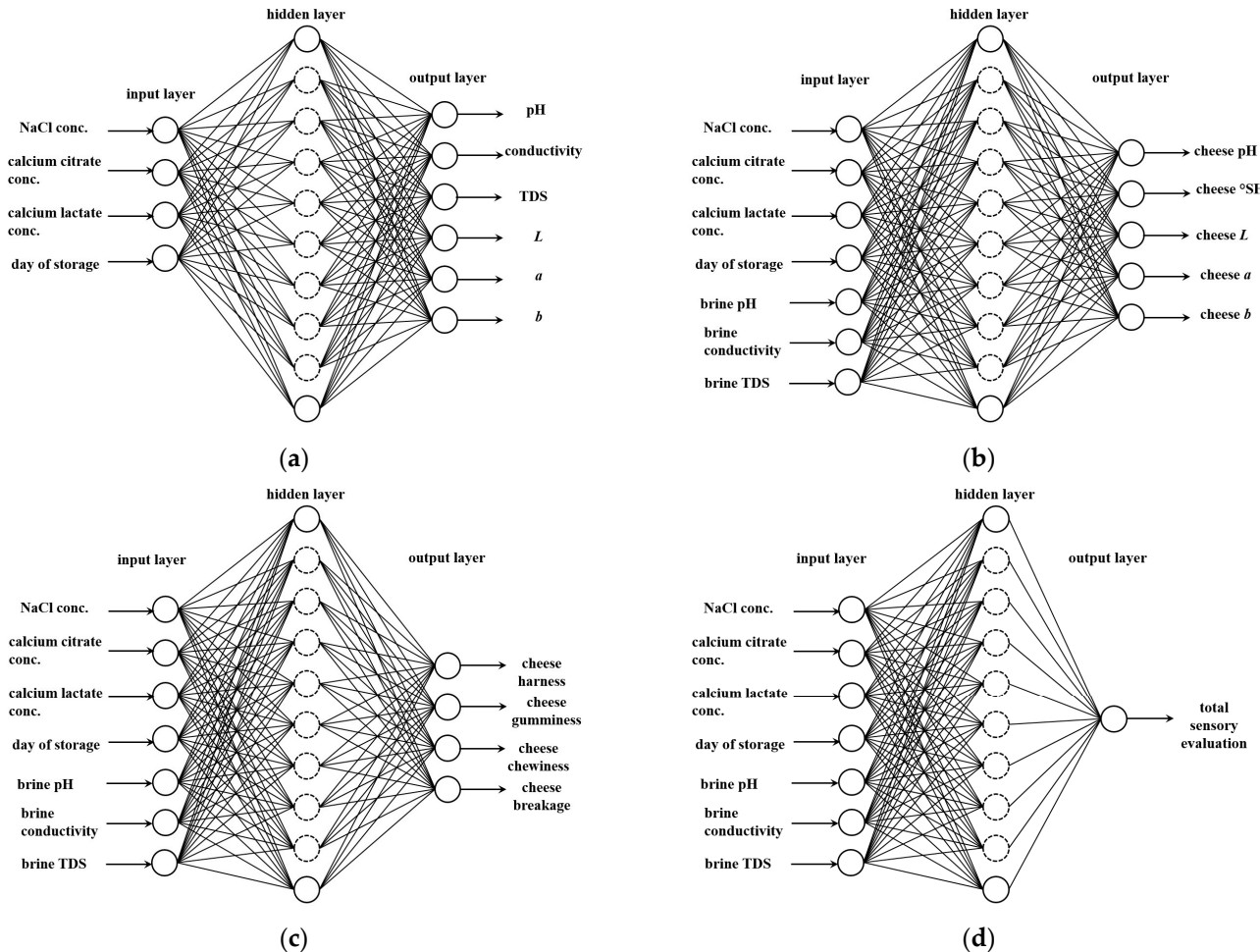

**Figure 1.** Architecture of ANN networks used for the prediction of (**a**) the properties of brine, (**b**) the physical properties of cheese, (**c**) the textural properties of cheese, (**d**) the total sensory evolution of cheese.

## 3. Results and Discussion

### 3.1. Physicochemical Properties of Brine and Cheese

Table 2 shows the physicochemical characteristics of the control brine (BC) and brines prepared with substitutional salts (Ca-citrate (BCC1, BCC2) and Ca-lactate (BCL1, BCL2)) during the 28 days of cold storage. The pH value of the control brine (BC) did not significantly change until the 7th day of storage, and at the end of the storage period, it had significantly increased (from 4.70 to 4.99 pH). The pH value of brines with substitutional salts (BCC1, BCC2, BCL1, BCL2), unlike the control brine, increased significantly on the 7th day of storage, and the increasing trend was noticeable until the end of the storage period in all brines. The highest pH value was measured in brine BCC1 (Ca-citrate 25%), which amounted to 5.19 pH units. The greatest significant difference between brines was measured on the 21st day of cold storage. Generally, the conductivity depends on the NaCl concentration in the solution; a higher NaCl concentration is proportional to higher conductivity values [32]. Brine conductivity changes within different NaCl concentrations. The highest conductivity was present before cheese ripening in the control brine and was 104.80 mS. During the storage period and cheese ripening period, a significant conductivity decrease occurred in the control and other samples as well. The explanation for this lies in the migration of NaCl from the brine into the cheese. Furthermore, conductivity values were much lower in brines with substitutional salts compared to the control brine during the storage period. Also, when 50% of NaCl was replaced with other salts, the conductivity was lower compared to in the 25% NaCl replacement. The total dissolved solids (TDS)

present the total concentration of dissolved solids in the tested liquid. All brine types were prepared with the same amount of salt, and the TDS values were different in different brine types. Control brine (BC) had the highest amount of TDS, and the amount decreased during the storage period from 52.70 to 33.10 g/L. The reason for the TDS decrease was the migration of salt into the cheese. The same trend was observed for all brine types with substitutional salts (BCC1, BCC2, BCL1, BCL2). According to the TDS values, NaCl has better dissolving properties than Ca-citrate or Ca-lactate. To confirm this statement, when Ca-citrate and Ca-lactate brine was prepared, after pasteurization and the cooling of brine, precipitation was visible at the bottom of the glass jar.

**Table 2.** Mean values for physical characteristics (pH value, conductivity (S), total dissolved solids (TDS), and color (L*, a*, b*) of brine during white brined cheese storage at 4 °C for 28 days.

| Variable | Days of Storage | Sample | | | | |
|---|---|---|---|---|---|---|
| | | BC | BCC1 | BCC2 | BCL1 | BCL2 |
| pH | 0 | 4.70 ± 0.00 [A,a] | 4.70 ± 0.00 [A,a] | 4.70 ± 0.00 [A,a] | 4.70 ± 0.00 [A,a] | 4.70 ± 0.00 [A,a] |
| | 7 | 4.77 ± 0.02 [A,a] | 4.82 ± 0.03 [A,b] | 4.90 ± 0.03 [A,b] | 4.95 ± 0.01 [B,b] | 4.96 ± 0.04 [C,b] |
| | 14 | 4.88 ± 0.09 [A,b] | 4.98 ± 0.03 [A,c] | 4.97 ± 0.02 [A,c] | 5.02 ± 0.01 [B,c] | 4.99 ± 0.04 [A,c] |
| | 21 | 4.91 ± 0.06 [A,c] | 4.98 ± 0.00 [A,c] | 5.08 ± 0.07 [B,d] | 5.09 ± 0.02 [C,d] | 5.08 ± 0.02 [D,d] |
| | 28 | 4.99 ± 0.06 [A,d] | 5.19 ± 0.11 [B,d] | 4.93 ± 0.04 [A,e] | 4.97 ± 0.02 [A,e] | 4.90 ± 0.05 [A,e] |
| S (mS) | 0 | 104.80 ± 2.89 [A,a] | 72.50 ± 0.42 [B,a] | 48.30 ± 0.49 [C,a] | 62.50 ± 0.00 [D,a] | 42.60 ± 0.51 [E,a] |
| | 7 | 87.30 ± 2.53 [A,b] | 61.60 ± 0.57 [B,b] | 41.90 ± 0.28 [C,b] | 55.00 ± 0.64 [D,b] | 38.80 ± 0.24 [E,a] |
| | 14 | 84.70 ± 2.48 [A,c] | 52.30 ± 0.14 [B,c] | 38.40 ± 0.31 [C,c] | 51.55 ± 0.21 [D,c] | 40.30 ± 0.62 [E,a] |
| | 21 | 80.50 ± 5.87 [A,d] | 55.30 ± 0.49 [B,d] | 40.50 ± 0.48 [C,d] | 51.20 ± 0.14 [D,d] | 38.50 ± 0.55 [E,a] |
| | 28 | 67.20 ± 3.35 [A,e] | 48.00 ± 1.48 [B,e] | 36.00 ± 0.89 [C,e] | 49.70 ± 0.85 [D,e] | 37.10 ± 0.38 [E,b] |
| TDS (g/L) | 0 | 52.70 ± 1.50 [A,a] | 36.05 ± 0.07 [B,a] | 24.10 ± 0.12 [C,a] | 31.20 ± 0.00 [D,a] | 21.60 ± 0.22 [E,a] |
| | 7 | 44.30 ± 0.49 [A,a] | 30.60 ± 0.71 [B,a] | 21.10 ± 0.24 [C,b] | 26.00 ± 0.85 [D,b] | 20.80 ± 0.15 [E,a] |
| | 14 | 42.70 ± 1.20 [A,a] | 25.90 ± 0.28 [B,a] | 21.40 ± 0.11 [C,a] | 25.80 ± 0.00 [D,c] | 20.50 ± 0.07 [E,a] |
| | 21 | 39.80 ± 2.99 [A,a] | 26.70 ± 0.92 [B,a] | 20.10 ± 0.32 [C,c] | 25.20 ± 0.14 [D,d] | 18.90 ± 0.10 [E,a] |
| | 28 | 33.10 ± 2.02 [A,a] | 24.00 ± 0.49 [B,a] | 19.90 ± 0.31 [C,d] | 24.80 ± 0.49 [D,e] | 17.20 ± 0.23 [E,b] |
| L* | 0 | 99.84 ± 0.11 [A,a] | 98.50 ± 0.00 [A,a] | 98.31 ± 1.28 [A,a] | 99.95 ± 0.00 [A,a] | 99.96 ± 1.03 [A,a] |
| | 7 | 96.77 ± 0.13 [A,a] | 83.89 ± 1.05 [B,b] | 60.82 ± 0.89 [C,b] | 93.89 ± 0.54 [A,a] | 95.42 ± 2.15 [A,b] |
| | 14 | 91.94 ± 1.77 [A,a] | 80.11 ± 6.83 [B,c] | 48.48 ± 3.24 [C,c] | 92.01 ± 3.11 [A,a] | 92.59 ± 0.99 [A,c] |
| | 21 | 92.64 ± 0.42 [A,a] | 71.56 ± 4.08 [B,d] | 44.48 ± 2.23 [C,d] | 91.08 ± 4.56 [A,a] | 76.35 ± 3.25 [D,a] |
| | 28 | 92.36 ± 0.60 [A,a] | 45.55 ± 10.39 [B,e] | 39.36 ± 3.05 [C,e] | 71.01 ± 6.87 [D,b] | 46.69 ± 5.11 [E,d] |
| a* | 0 | 0.03 ± 0.00 [A,a] | 0.13 ± 0.00 [A,a] | 0.15 ± 0.02 [A,a] | 0.02 ± 0.00 [A,a] | 0.01 ± 0.01 [A,a] |
| | 7 | −0.31 ± 0.04 [A,a] | 0.86 ± 0.00 [B,b] | 1.41 ± 0.02 [C,b] | 0.13 ± 0.04 [A,a] | −0.05 ± 0.03 [A,a] |
| | 14 | −0.12 ± 0.25 [A,a] | 0.94 ± 0.36 [B,c] | −0.01 ± 0.11 [A,a] | 0.08 ± 0.21 [A,a] | 0.01 ± 0.13 [A,a] |
| | 21 | −0.24 ± 0.07 [A,a] | 1.29 ± 0.06 [B,d] | −0.42 ± 0.05 [A,c] | 0.10 ± 0.31 [A,a] | 0.65 ± 0.09 [A,b] |
| | 28 | −0.12 ± 0.03 [A,a] | −0.12 ± 0.16 [A,a] | −0.64 ± 0.07 [B,d] | 0.61 ± 0.08 [C,b] | 0.66 ± 0.23 [D,c] |

**Table 2.** *Cont.*

| Variable | Days of Storage | Sample | | | | |
|---|---|---|---|---|---|---|
| | | BC | BCC1 | BCC2 | BCL1 | BCL2 |
| b* | 0 | 0.08 ± 0.02 [A,a] | 0.71 ± 0.00 [A,a] | 0.76 ± 0.03 [A,a] | 0.06 ± 0.00 [A,a] | 0.09 ± 0.01 [A,a] |
| | 7 | 3.77 ± 0.12 [A,a] | 6.65 ± 0.16 [A,a] | 6.75 ± 0.15 [A,b] | 3.97 ± 0.11 [A,a] | 3.66 ± 0.22 [A,a] |
| | 14 | 4.93 ± 0.24 [A,a] | 7.41 ± 1.01 [A,b] | 7.17 ± 0.37 [A,c] | 4.79 ± 0.56 [A,a] | 4.71 ± 0.63 [A,a] |
| | 21 | 5.94 ± 0.30 [A,a] | 8.44 ± 0.09 [A,c] | 10.23 ± 0.21 [A,d] | 5.24 ± 0.85 [A,a] | 7.10 ± 0.49 [A,b] |
| | 28 | 6.33 ± 0.03 [A,b] | 14.33 ± 9.26 [B,d] | 15.79 ± 0.14 [C,e] | 7.84 ± 0.01 [A,b] | 12.35 ± 0.11 [D,c] |

BC—control brine with 100% NaCl; BCC1—brine with 75% NaCl and 25% Ca-citrate; BCC2—brine with 50% NaCl and 50% Ca-citrate; BCL1—brine with 75% NaCl and 25% Ca-lactate; BCL2—brine with 50% NaCl and 50% Ca-lactate. [A–E] The same superscript capital letters within a row denote no significant differences ($p > 0.05$) between the values obtained for the different trials regarding the control sample according to Tukey's ANOVA. [a–e] The same superscript lowercase letters within a column denote no significant differences ($p > 0.05$) between values obtained for different days of storage according to Tukey's ANOVA.

None of the three color coordinates (L*, a*, b*) significantly differed over the storage period in the control brine (Table 2). The control brine (BC) color was stable over the cheese ripening period. When substitutional salt brines (BCC1, BCC2, BCL1, BCL2) were observed, the color significantly changed over the storage period. The L* value significantly decreased. Meanwhile, the a* and b* values significantly increased. There was also a visible change between brines, whereas the control brine was transparent over the storage period, and substitutional salt brines were white, not transparent.

Table 3 shows the characteristics of the produced white brined cheeses (pH value, titratable acidity (°SH), color (L*, a*, b*)) during the storage period. Over the storage period, the pH value of the control cheese significantly increased from 4.85 to 5.23. The pH value of cheese ripened over substitutional salt brine was stable throughout the whole storage period and did not significantly differ among white brined cheeses. Titratable acidity is determined by neutralizing the acid present in a known quantity (weight or volume) of a food sample using a standard base and is expressed as the Soxhlet–Henkel degree (°SH) [10]. Control cheese brined in NaCl brine had the highest titratable acidity after production, and this significantly decreased over the storage period. The same trend was observed in cheeses brined in Ca-citrate (CC1, CC2) and Ca-lactate (CL1, CL2) brines.

**Table 3.** Mean values for the physical characteristics (pH value, titratable acidity (°SH), color (L*, a*, b*)) of white brined cheese during storage at 4 °C for 28 days.

| Variable | Days of Storage | Sample | | | | |
|---|---|---|---|---|---|---|
| | | C | CC1 | CC2 | CL1 | CL2 |
| pH | 0 | 4.85 ± 0.10 [A,a] | 5.04 ± 0.00 [A,a] | 5.13 ± 0.03 [B,a] | 5.13 ± 0.00 [A,a] | 5.13 ± 0.05 [C,a] |
| | 7 | 4.87 ± 0.13 [A,a] | 5.13 ± 0.01 [B,a] | 4.94 ± 0.02 [A,a] | 5.05 ± 0.09 [A,a] | 5.00 ± 0.11 [A,a] |
| | 14 | 5.00 ± 0.11 [A,a] | 5.22 ± 0.11 [A,a] | 5.10 ± 0.05 [A,a] | 5.09 ± 0.08 [A,a] | 5.08 ± 0.03 [A,a] |
| | 21 | 5.13 ± 0.11 [A,b] | 5.14 ± 0.03 [A,a] | 5.16 ± 0.11 [A,a] | 5.19 ± 0.03 [A,a] | 5.18 ± 0.07 [A,a] |
| | 28 | 5.23 ± 0.10 [A,c] | 5.25 ± 0.08 [A,a] | 5.06 ± 0.10 [A,e] | 5.09 ± 0.01 [A,a] | 5.08 ± 0.07 [A,a] |
| °SH | 0 | 83.20 ± 0.00 [A,a] | 63.20 ± 0.00 [B,a] | 79.20 ± 0.21 [A,a] | 79.20 ± 0.00 [A,a] | 79.20 ± 0.01 [A,a] |
| | 7 | 46.50 ± 7.58 [A,b] | 27.75 ± 0.78 [B,b] | 30.40 ± 0.17 [C,b] | 35.20 ± 0.00 [D,b] | 34.40 ± 0.15 [E,b] |
| | 14 | 34.80 ± 5.41 [A,c] | 26.60 ± 0.64 [B,c] | 24.00 ± 0.32 [C,c] | 39.60 ± 0.57 [A,c] | 38.40 ± 0.19 [A,c] |
| | 21 | 30.00 ± 4.00 [A,d] | 29.00 ± 1.41 [A,d] | 27.20 ± 0.14 [A,d] | 41.20 ± 0.57 [B,d] | 35.20 ± 0.31 [C,d] |
| | 28 | 35.90 ± 1.44 [A,e] | 22.40 ± 5.66 [B,e] | 33.60 ± 1.13 [A,e] | 32.00 ± 1.13 [A,e] | 34.40 ± 0.78 [A,e] |

**Table 3.** *Cont.*

| Variable | Days of Storage | Sample | | | | |
|---|---|---|---|---|---|---|
| | | C | CC1 | CC2 | CL1 | CL2 |
| L* | 0 | 93.52 ± 0.93 [A,a] | 93.52 ± 0.93 [A,a] | 93.52 ± 0.93 [A,a] | 93.52 ± 0.93 [A,a] | 93.52 ± 0.93 [A,a] |
| | 7 | 94.21 ± 3.08 [A,a] | 91.76 ± 0.59 [A,a] | 93.04 ± 0.88 [A,a] | 92.04 ± 0.40 [A,a] | 93.32 ± 1.48 [A,a] |
| | 14 | 91.57 ± 2.55 [A,a] | 92.31 ± 0.07 [A,a] | 93.25 ± 0.62 [A,a] | 94.04 ± 0.32 [A,a] | 92.85 ± 0.76 [A,a] |
| | 21 | 91.80 ± 1.91 [A,a] | 93.38 ± 0.05 [A,a] | 91.48 ± 1.24 [A,a] | 93.36 ± 0.34 [A,a] | 92.99 ± 1.05 [A,a] |
| | 28 | 92.45 ± 0.13 [A,a] | 93.12 ± 0.37 [A,a] | 91.87 ± 0.99 [A,a] | 92.37 ± 2.01 [A,a] | 91.90 ± 0.27 [A,a] |
| a* | 0 | −0.49 ± 0.40 [A,a] | −0.49 ± 0.40 [A,a] | −0.49 ± 0.40 [A,a] | −0.49 ± 0.40 [A,a] | −0.49 ± 0.40 [A,a] |
| | 7 | −0.33 ± 0.67 [A,a] | −1.08 ± 0.05 [A,a] | −0.90 ± 0.28 [A,a] | −0.55 ± 0.35 [A,a] | −0.79 ± 0.07 [A,a] |
| | 14 | −0.47 ± 0.43 [A,a] | −0.50 ± 0.08 [A,a] | −0.59 ± 0.17 [A,a] | −0.61 ± 0.01 [A,a] | −0.59 ± 0.34 [A,a] |
| | 21 | −0.50 ± 0.25 [A,a] | −0.31 ± 0.27 [A,a] | −1.05 ± 0.28 [A,a] | −0.47 ± 0.12 [A,a] | −1.05 ± 0.28 [A,a] |
| | 28 | −0.93 ± 0.43 [A,a] | −0.10 ± 0.05 [A,a] | −0.84 ± 0.21 [A,a] | −0.26 ± 0.03 [A,a] | −0.84 ± 0.27 [A,a] |
| b* | 0 | 13.68 ± 1.46 [A,a] | 13.68 ±1.46 [A,a] | 13.68 ± 1.46 [A,a] | 13.68 ± 1.46 [A,a] | 13.68 ± 1.46 [A,a] |
| | 7 | 9.76 ± 2.60 [A,a] | 12.31 ± 0.54 [A,a] | 11.71 ± 1.09 [A,a] | 11.38 ± 1.24 [A,a] | 12.53 ± 1.92 [A,a] |
| | 14 | 13.11 ± 2.18 [A,a] | 12.95 ± 0.76 [A,a] | 10.81 ± 0.99 [A,a] | 11.39 ± 0.15 [A,a] | 12.32 ± 1.35 [A,a] |
| | 21 | 11.71 ± 1.69 [A,a] | 11.44 ± 1.04 [A,a] | 12.69 ± 1.15 [A,a] | 11.64 ± 0.43 [A,a] | 11.01 ± 2.01 [A,a] |
| | 28 | 10.46 ± 0.95 [A,a] | 11.41 ± 0.35 [A,a] | 12.01 ± 0.77 [A,a] | 11.90 ± 1.88 [A,a] | 13.13 ± 1.88 [A,a] |

C—control cheese from brine with 100% NaCl; CC1—cheese from brine with 75% of NaCl and 25%. Ca-citrate; CC2—cheese from brine with 50% of NaCl and 50% Ca-citrate; CL1—cheese from brine with 75% of NaCl and 25% Ca-lactate; CL2—cheese from brine with 50% of NaCl and 50% Ca-lactate [A–E] The same superscript capital letters within a row denote no significant differences ($p > 0.05$) between the values obtained for the different trials regarding the control sample according to Tukey's ANOVA. [a–e] The same superscript lowercase letters within a column denote no significant differences ($p > 0.05$) between values obtained for different days of storage according to Tukey's ANOVA.

The color coordinates L*, a*, and b* did not significantly differ among the control cheese and other cheeses, and they were the same during the 28 days of the storage period.

Figure 2 presents the sodium chloride content in cheese determined by Mohr's method. The control sample (C) had the highest NaCl content, 5.28%, after seven days of cold storage. After 14 days, the NaCl amount in the cheese increased to 6.27%, whereas on the 21st day, a drop in NaCl was measured, leading to a content of 5.65%. At the end of the storage period, the control cheese had 6.20% NaCl. A similar trend was observed in the other cheeses. Cheeses brined in Ca-citrate brine with 25% (CC1) and 50% (CC2) replacements had lower NaCl amounts compared to the control cheese after seven days: 3.67%, and 3.25% NaCl, respectively. Over the storage period, the NaCl amount increased in both CC1 and CC2 samples, and it was 5.34% and 4.18%, respectively, at the end of the storage period. Cheeses brined in Ca-lactate brine with 25% (CL1) and 50% (CL2) replacements had lower NaCl amounts compared to the control cheese after seven days: 3.76%, and 2.74% NaCl, respectively. Over the storage period, the NaCl amount increased in both CL1 and CL2, and at the end of the storage period, it was 5.02% and 3.77%, respectively. The salt increase in all cheese samples was dueto NaCl migration from brine to cheese. The greatest reduction in NaCl in cheese at the end of the storage period was observed in cheese brined in Ca-lactate with 50% NaCl replacement and was around 40%. Katsiari et al. [17] reported similar results where they achieved a 50% NaCl reduction when they prepared brine with a 1:1 ratio of NaCl:KCl.

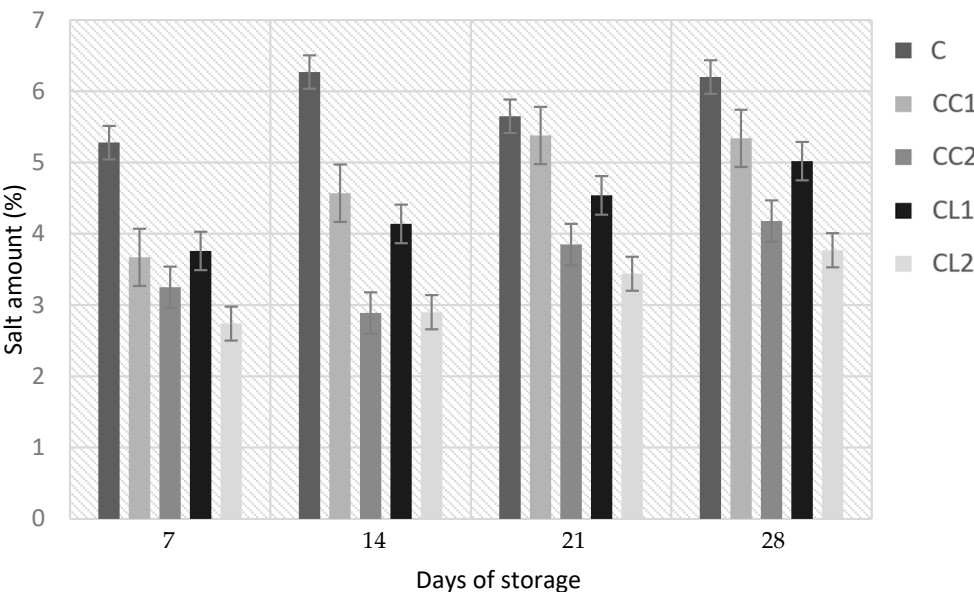

**Figure 2.** Sodium chloride amount in the control cheese (C), cheese ripened in Ca-citrate brine (CC1, CC2) and Ca-lactate (CL1, CL2). C—control cheese from brine with 100% NaCl; CC1—cheese from brine with 75% of NaCl and 25% Ca-citrate; CC2—cheese from brine with 50% of NaCl and 50% Ca-citrate; CL1—cheese from brine with 75% of NaCl and 25% Ca-lactate; CL2—cheese from brine with 50% NaCl and 50% Ca-lactate. Error bars show standard deviations (*n* = 3).

### 3.2. Textural Properties

Sodium chloride, among its many roles in the cheese production process, influences cheese's textural properties. Table 4 shows the cheeses' textural properties, hardness (N), adhesive force (N), adhesiveness (Nmm), cohesiveness (N/m), gumminess (N), postponed elasticity (mm), chewiness (Nmm), resistance, breakage (N), and fibrousness (mm). Generally, in all cheese samples, hardness decreases over the storage period, although a statistically significant difference was only observed in control cheese (C) (*p* < 0.05). Control cheese samples had the highest hardness after seven days compared to other cheese samples ripened in substitutional salt brines. This behavior was statistically significant on every analysis day, except on the 14th day. The highest hardness was measured in the control sample (C) after seven days of storage, and it amounted to 10.35 N, and the lowest value occurred in the sample brined in 50% NaCl and 50% Ca-lactate after 28 days of analysis and was 2.46 N. Thibadeau et al. [33] produced brine-salted mozzarella-type cheeses with different ratios of NaCl and KCl (100% NaCl; 25% KCl and 75% NaCl; 50% KCl and 50% NaCl in brine). They concluded that the K content maintained the hardness at the surface of melted cheese over a storage period of 28 days at 4 °C. Furthermore, Katsiari et al. [17], who produced feta cheese brined in different ratios of NaCl and KCl (3:1 and 1:1, respectively) and conducted analyses after 60, 120, and 240 days of brining, concluded that the cheeses salted with the NaCl/KCl mixtures were generally more fracturable and were softer. Their results are in correspondence with the results obtained in this research, regardless of the different replacement salts used. According to the results, a reduced NaCl amount results in cheese with a lower hardness.

**Table 4.** Mean values for the textural properties of white brined cheese during storage at 4 °C for 28 days.

| Textural Property | Days of Storage | Sample | | | | |
|---|---|---|---|---|---|---|
| | | C | CC1 | CC2 | CL1 | CL2 |
| Hardness (N) | 7 | 10.35 ± 3.26 [A,a] | 6.65 ± 0.71 [A,a] | 5.25 ± 0.59 [B,a] | 5.18 ± 1.21 [C,a] | 3.77 ± 0.31 [D,a] |
| | 14 | 8.18 ± 3.39 [A,a] | 5.97 ± 1.61 [A,a] | 4.15 ± 0.60 [A,a] | 4.34 ± 0.67 [A,a] | 3.93 ± 0.76 [A,a] |
| | 21 | 8.64 ± 3.13 [A,a] | 4.04 ± 0.43 [B,a] | 2.72 ± 0.02 [C,a] | 3.41 ± 0.14 [D,a] | 3.09 ± 0.38 [E,a] |
| | 28 | 5.21 ± 0.80 [A,b] | 3.42 ± 0.56 [A,a] | 4.31 ± 2.09 [A,a] | 3.15 ± 0.43 [A,a] | 2.46 ± 0.05 [A,a] |
| Adhesive force (N) | 7 | −0.15 ± 0.04 [A,a] | −0.17 ± 0.02 [A,a] | −0.16 ± 0.02 [A,a] | −0.13 ± 0.10 [A,a] | −0.06 ± 0.03 [A,a] |
| | 14 | −0.11 ± 0.05 [A,a] | −0.17 ± 0.08 [A,a] | −0.17 ± 0.02 [A,a] | −0.12 ± 0.03 [A,a] | −0.11 ± 0.03 [A,a] |
| | 21 | −0.16 ± 0.06 [A,a] | −0.10 ± 0.04 [A,a] | −0.17 ± 0.09 [A,a] | −0.09 ± 0.02 [A,a] | −0.12 ± 0.00 [A,a] |
| | 28 | −0.13 ± 0.04 [A,a] | −0.11 ± 0.02 [A,a] | −0.47 ± 0.50 [A,a] | −0.10 ± 0.03 [A,a] | −0.10 ± 0.00 [A,a] |
| Adhesiveness (Nmm) | 7 | 0.51 ± 0.11 [A,a] | 0.76 ± 0.33 [A,a] | 0.45 ± 0.15 [A,a] | 0.30 ± 0.25 [A,a] | 0.25 ± 0.16 [B,a] |
| | 14 | 0.44 ± 0.45 [A,a] | 0.60 ± 0.20 [A,a] | 0.59 ± 0.14 [A,a] | 0.65 ± 0.20 [A,a] | 0.46 ± 0.23 [A,a] |
| | 21 | 0.53 ± 0.07 [A,a] | 0.38 ± 0.13 [A,a] | 0.64 ± 0.14 [A,a] | 0.34 ± 0.05 [A,a] | 0.49 ± 0.16 [A,a] |
| | 28 | 0.46 ± 0.19 [A,a] | 0.59 ± 0.26 [A,a] | 0.53 ± 0.41 [A,a] | 0.47 ± 0.19 [A,a] | 0.31 ± 0.07 [A,a] |
| Cohesiveness (N/m) | 7 | 0.28 ± 0.05 [A,a] | 0.25 ± 0.04 [A,a] | 0.25 ± 0.02 [A,a] | 0.29 ± 0.02 [A,a] | 0.25 ± 0.02 [A,a] |
| | 14 | 0.24 ± 0.03 [A,a] | 0.20 ± 0.02 [A,a] | 0.22 ± 0.05 [A,a] | 0.24 ± 0.03 [A,a] | 0.28 ± 0.05 [A,a] |
| | 21 | 0.25 ± 0.02 [A,a] | 0.27 ± 0.01 [A,a] | 0.27 ± 0.01 [A,a] | 0.21 ± 0.00 [A,a] | 0.24 ± 0.04 [A,a] |
| | 28 | 0.22 ± 0.02 [A,a] | 0.21 ± 0.04 [A,a] | 0.32 ± 0.08 [A,a] | 0.25 ± 0.04 [A,a] | 0.28 ± 0.04 [A,a] |
| Gumminess (N) | 7 | 3.00 ± 1.27 [A,a] | 1.65 ± 0.21 [A,a] | 0.80 ± 0.15 [B,a] | 1.52 ± 0.42 [A,a] | 0.94 ± 0.10 [C,a] |
| | 14 | 1.98 ± 0.95 [A,a] | 1.19 ± 0.36 [A,a] | 1.26 ± 0.16 [A,a] | 1.04 ± 0.16 [A,a] | 1.14 ± 0.39 [A,a] |
| | 21 | 2.14 ± 0.71 [A,a] | 1.09 ± 0.09 [A,a] | 0.72 ± 0.03 [A,a] | 0.71 ± 0.02 [A,a] | 0.75 ± 0.21 [A,a] |
| | 28 | 1.17 ± 0.27 [A,b] | 0.74 ± 0.26 [A,a] | 1.46 ± 1.00 [A,a] | 0.81 ± 0.21 [A,a] | 0.69 ± 0.12 [A,a] |
| Postponed elasticity (mm) | 7 | −3.01 ± 2.28 [A,a] | −4.99 ± 0.51 [A,a] | −5.27 ± 1.26 [A,a] | −4.13 ± 0.99 [A,a] | −3.12 ± 1.94 [A,a] |
| | 14 | −3.70 ± 1.02 [A,a] | −4.88 ± 1.99 [A,a] | −4.13 ± 0.91 [A,a] | −4.34 ± 2.37 [A,a] | −4.27 ± 0.98 [A,a] |
| | 21 | −0.99 ± 1.09 [A,a] | −4.90 ± 0.13 [A,a] | −5.15 ± 0.07 [B,a] | −4.62 ± 1.83 [A,a] | −5.24 ± 0.54 [C,a] |
| | 28 | −5.69 ± 0.98 [A,a] | −5.21 ± 1.22 [A,a] | −3.02 ± 1.82 [A,a] | −5.27 ± 0.66 [A,a] | −4.49 ± 0.10 [A,a] |
| Chewiness (Nmm) | 7 | 15.92 ± 8.47 [A,a] | 5.05 ± 1.30 [B,a] | 3.28 ± 0.80 [C,a] | 3.28 ± 0.90 [D,a] | 1.90 ± 0.50 [E,a] |
| | 14 | 7.89 ± 5.46 [A,a] | 3.21 ± 1.60 [A,a] | 3.10 ± 0.06 [A,a] | 2.48 ± 0.63 [A,a] | 2.07 ± 1.04 [A,a] |
| | 21 | 10.73 ± 5.03 [A,B] | 2.07 ± 0.37 [B,a] | 0.94 ± 0.04 [C,a] | 1.02 ± 0.11 [D,a] | 1.05 ± 0.78 [E,a] |
| | 28 | 3.00 ± 1.08 [A,a] | 1.21 ± 0.79 [B,a] | 1.63 ± 1.09 [C,a] | 1.31 ± 0.62 [D,a] | 0.61 ± 0.09 [E,a] |
| Resistance | 7 | 0.27 ± 0.07 [A,a] | 0.21 ± 0.04 [A,a] | 0.22 ± 0.03 [A,a] | 0.23 ± 0.03 [A,a] | 0.34 ± 0.08 [A,a] |
| | 14 | 0.24 ± 0.07 [A,a] | 0.22 ± 0.11 [A,a] | 0.26 ± 0.02 [A,a] | 0.27 ± 0.13 [A,a] | 0.24 ± 0.04 [A,a] |
| | 21 | 0.38 ± 0.04 [A,a] | 0.22 ± 0.02 [A,a] | 0.23 ± 0.02 [A,a] | 0.25 ± 0.07 [A,a] | 0.21 ± 0.03 [A,a] |
| | 28 | 0.18 ± 0.02 [A,a] | 0.20 ± 0.05 [A,a] | 0.25 ± 0.05 [A,a] | 0.22 ± 0.05 [A,a] | 0.25 ± 0.04 [A,a] |
| Breakage (N) | 7 | 10.05 ± 3.35 [A,a] | 6.06 ± 0.79 [A,a] | 4.58 ± 0.02 [B,a] | 4.54 ± 0.99 [C,a] | 3.33 ± 0.70 [D,a] |
| | 14 | 7.37 ± 2.80 [A,a] | 5.11 ± 2.11 [A,a] | 3.28 ± 0.57 [A,a] | 3.68 ± 0.70 [A,a] | 3.04 ± 0.98 [B,a] |
| | 21 | 8.46 ± 3.05 [A,a] | 3.16 ± 0.27 [B,a] | 2.14 ± 0.02 [C,a] | 2.99 ± 0.62 [D,a] | 2.47 ± 0.55 [E,a] |
| | 28 | 4.58 ± 0.57 [A,b] | 2.63 ± 0.44 [A,a] | 2.67 ± 0.22 [A,a] | 2.89 ± 0.43 [A,a] | 2.26 ± 0.00 [A,a] |

**Table 4.** *Cont.*

| Textural Property | Days of Storage | Sample | | | | |
|---|---|---|---|---|---|---|
| | | **C** | **CC1** | **CC2** | **CL1** | **CL2** |
| Fibrousness (mm) | 7 | 8.04 ± 2.55 [A,a] | 7.99 ± 2.95 [A,a] | 7.35 ± 3.52 [A,a] | 6.36 ± 2.48 [A,a] | 7.79 ± 1.77 [A,a] |
| | 14 | 7.43 ± 4.40 [A,a] | 10.72 ± 2.21 [A,a] | 6.59 ± 3.24 [A,a] | 7.40 ± 2.74 [A,a] | 12.91 ± 1.22 [A,a] |
| | 21 | 5.61 ± 1.47 [A,a] | 7.47 ± 3.11 [A,a] | 8.63 ± 3.35 [A,a] | 5.39 ± 0.48 [A,a] | 6.01 ± 2.66 [A,a] |
| | 28 | 8.69 ± 3.00 [A,a] | 9.23 ± 4.47 [A,a] | 3.63 ± 1.71 [A,a] | 8.85 ± 2.11 [A,a] | 7.17 ± 4.12 [A,a] |

C—control cheese from brine with 100% NaCl; CC1—cheese from brine with 75% of NaCl and 25% Ca-citrate; CC2—cheese from brine with 50% NaCl and 50% Ca-citrate; CL1—cheese from brine with 75% NaCl and 25% Ca-lactate; CL2—cheese from brine with 50% NaCl and 50% Ca-lactate [A–E] The same superscript capital letters within a row denote no significant differences ($p > 0.05$) between the values obtained for the different trials regarding the control sample according to Tukey's ANOVA. [a–b] The same superscript lowercase letters within a column denote no significant differences ($p > 0.05$) between values obtained for different days of storage according to Tukey's ANOVA.

Other textural properties (adhesive force (N), adhesiveness (Nmm), cohesiveness (N/m), gumminess (N), postponed elasticity (mm), chewiness (Nmm), resistance, breakage (N) and fibrousness (mm)) determined in control cheese samples and cheese produced with substitutional salts were not statistically significant ($p > 0.05$) over the storage period and among the samples, although the numerical values were different for some properties. The greatest differences among the samples and days of storage occurred for gumminess, chewiness, and breakage. All three properties were the highest in the control sample after seven days of ripening in brine (3.00 N, 15.92 Nmm, and 10.05 N) and the lowest values occurred in the sample brined in 50% NaCl and 50% Ca-lactate after 28 days (0.69 N, 0.61 Nmm, and 2.26 N). Ripening in Ca-lactate resulted in a softer and more breakable cheese compared to the control and Ca-citrate ripened cheeses. Katsiari et al. [17] reported that there were no significant differences ($p > 0.05$) in the force and compression required to fracture the control sample and the samples brined in different ratios of NaCl and KCl (3:1 and 1:1).

Some textural properties of the produced white brined cheese can be linked to its physicochemical characteristics. According to the literature, higher pH levels, protein contents, and total solids contents generally result in harder cheeses with a decreased deformation capacity [10]. However, our results show the opposite behavior. The pH values of the control cheese increase over the storage period, and hardness decreases. This contrasting outcome could be attributed to the ripening process occurring in the brine, where additional mechanisms take effect.

Salt impacts curd syneresis by creating a combined effect of the osmotic movement of water molecules in response to the salt gradient as well as the contraction or expansion of the curd protein network due to the salt concentration [34]. Lu and McMahon [34] found in their research that, at low salt ratios, the curd protein solubility increased, leading to the expansion of the protein matrix. Conversely, at high salt ratios, the protein solubility decreased, causing dehydration and the contraction of the curd protein matrix in cheddar cheese.

Furthermore, during the ripening period, casein hydrolysis or the dissolution of some of the residual colloidal calcium phosphate in the casein matrix of the cheese could have contributed to decreases in the hardness values of all cheese samples [35].

### 3.3. Sensory Properties

The results of the sensory evaluation of the cheese samples, the including appearance, color, consistency, cut, odor, taste, and total sensory score after the 7th, 14th, 21st, and 28th days of storage are presented in Table 5. According to the statistical analysis, individual properties of different cheese samples were not statistically significant after the 7th, 14th, 21st, and 28th days of analysis. There were no significant differences between samples in terms of the individual properties. The appearance of all tested samples ranged from

1.7 to 2.0 (out of max 2), and panellists did not notice a significant difference between control and replacement salt samples. The color of all samples was the characteristic white of feta-type cheese, and the panellists scored color from 0.80 to 1.0 (max of 1). The consistency of all cheese samples was scored between 1.60 and 2.0 (max of 2). The control sample had the highest score during the entire storage period and the lowest scores were obtained for cheese ripened in Ca-citrate (CC2) on the 28th day and cheese ripened in Ca-lactate (CL1, CL2) on the 21st day of analysis. Panellists explained the lower scores for named samples as being due to the softer cheese consistency. The sensory analysis scores are in correspondence with cheese's textural properties (Table 4). The cut and odor of all tested samples were more or less similar between samples and days of storage. The highest score for taste was obtained for the control sample (C) as well as for cheese brined in Ca-citrate (25% Ca-citrate). The lowest score (7.8 out of max 10) was obtained for sample CL2 (Ca-lactate, 50%) after the 21st day of analysis. Panellists explained the lower taste score as being due to the bitter taste presence and gritty texture in the mouth. The statistical analysis of the total sensory score showed significance after the 7th, 14th, and 28th days. The control sample (C) obtained the highest total score (19.3, out of max 20) after 14 days of ripening in control brine, and the lowest total score (16.4, out of max 20) was obtained for the Ca-lactate (50%) sample after the 21st day of ripening. Ayyash et al. [36] observed that the flavor of Mozzarella cheeses salted with different mixtures of NaCl/KCl was not affected. Katsiari et al. [17] reported similar results for various cheese varieties. Hence, the limited partial replacement of NaCl with KCl did not significantly modify the cheeses' flavor. However, the substitution of KCl for NaCl should not be more than 25%. Above this substitution percentage, cheese can develop a metallic off-flavor [18,25], as observed in the present study.

**Table 5.** Mean values for the sensory properties of white brined cheese during storage at 4 °C for 28 days.

| Sensory Property | Days of Storage | Sample | | | | |
|---|---|---|---|---|---|---|
| | | C | CC1 | CC2 | CL1 | CL2 |
| Appearance (max. 2) | 7 | 1.90 ± 0.14 [A,a] | 2.00 ± 0.10 [A,a] | 1.90 ± 0.09 [A,a] | 1.90 ± 0.14 [A,a] | 1.90 ± 0.09 [A,a] |
| | 14 | 1.90 ± 0.11 [A,a] | 1.80 ± 0.28 [A,a] | 2.00 ± 0.11 [A,a] | 1.80 ± 0.40 [A,a] | 2.00 ± 0.11 [A,a] |
| | 21 | 1.90 ± 0.28 [A,a] | 1.90 ± 0.09 [A,a] | 1.70 ± 0.42 [A,a] | 1.80 ± 0.36 [A,a] | 1.70 ± 0.37 [A,a] |
| | 28 | 1.90 ± 0.17 [A,a] | 1.90 ± 0.12 [A,a] | 1.70 ± 0.34 [A,a] | 1.90 ± 0.09 [A,a] | 1.90 ± 0.08 [A,a] |
| Color (max. 1) | 7 | 1.00 ± 0.39 [A,a] | 1.00 ± 0.05 [A,a] | 0.90 ± 0.10 [A,a] | 1.00 ± 0.29 [A,a] | 1.00 ± 0.32 [A,a] |
| | 14 | 1.00 ± 0.05 [A,a] | 1.00 ± 1.38 [A,a] | 1.00 ± 0.05 [A,a] | 1.00 ± 0.71 [A,a] | 1.00 ± 0.05 [A,a] |
| | 21 | 1.00 ± 0.02 [A,a] | 0.90 ± 0.32 [A,a] | 0.80 ± 0.22 [A,a] | 0.90 ± 0.13 [A,a] | 0.90 ± 0.15 [A,a] |
| | 28 | 0.90 ± 0.15 [A,a] | 0.90 ± 0.11 [A,a] | 1.00 ± 0.34 [A,a] | 1.00 ± 0.28 [A,a] | 1.00 ± 0.35 [A,a] |
| Consistency (max. 2) | 7 | 1.90 ± 0.26 [A,a] | 1.90 ± 0.21 [A,a] | 1.80 ± 0.18 [A,a] | 1.90 ± 0.13 [A,a] | 1.90 ± 0.12 [A,a] |
| | 14 | 2.00 ± 0.02 [A,a] | 1.90 ± 0.21 [A,a] | 1.80 ± 0.38 [A,a] | 1.90 ± 0.15 [A,a] | 1.90 ± 0.17 [A,a] |
| | 21 | 1.90 ± 0.14 [A,a] | 2.00 ± 0.30 [A,a] | 1.70 ± 0.61 [A,a] | 1.60 ± 0.46 [A,a] | 1.60 ± 0.50 [A,a] |
| | 28 | 1.90 ± 0.24 [A,a] | 1.70 ± 0.41 [A,a] | 1.60 ± 0.28 [A,a] | 1.90 ± 0.12 [A,a] | 1.80 ± 0.11 [A,a] |
| Cut (max. 3) | 7 | 2.90 ± 0.84 [A,a] | 3.00 ± 0.10 [A,a] | 2.60 ± 0.82 [A,a] | 3.00 ± 0.00 [A,a] | 2.90 ± 0.18 [A,a] |
| | 14 | 3.00 ± 0.00 [A,a] | 2.90 ± 0.27 [A,a] | 3.00 ± 0.04 [A,a] | 2.70 ± 0.39 [A,a] | 3.00 ± 0.05 [A,a] |
| | 21 | 3.00 ± 0.05 [A,a] | 3.00 ± 0.07 [A,a] | 2.70 ± 0.55 [A,a] | 2.80 ± 0.26 [A,a] | 2.60 ± 0.60 [A,a] |
| | 28 | 2.80 ± 0.36 [A,a] | 2.80 ± 0.31 [A,a] | 2.40 ± 0.58 [A,a] | 2.00 ± 0.10 [A,a] | 2.70 ± 0.38 [A,a] |

**Table 5.** *Cont.*

| Sensory Property | Days of Storage | Sample | | | | |
|---|---|---|---|---|---|---|
| | | C | CC1 | CC2 | CL1 | CL2 |
| Odor (max. 2) | 7 | 2.00 ± 0.10 [A,a] | 1.90 ± 0.13 [A,a] | 2.0 ± 0.09 [A,a] | 2.00 ± 0.05 [A,a] | 1.90 ± 0.18 [A,a] |
| | 14 | 1.90 ± 0.27 [A,a] | 2.00 ± 0.09 [A,a] | 2.0 ± 0.07 [A,a] | 2.00 ± 0.10 [A,a] | 2.00 ± 0.07 [A,a] |
| | 21 | 2.00 ± 0.05 [A,a] | 1.90 ± 0.15 [A,a] | 1.90 ± 0.19 [A,a] | 1.90 ± 0.29 [A,a] | 1.80 ± 0.39 [A,a] |
| | 28 | 2.00 ± 0.04 [A,a] | 2.00 ± 0.07 [A,a] | 1.90 ± 0.13 [A,a] | 2.00 ± 0.06 [A,a] | 2.00 ± 0.11 [A,a] |
| Taste (max. 10) | 7 | 9.00 ± 0.97 [A,a] | 9.40 ± 0.80 [A,a] | 9.00 ± 0.70 [A,a] | 8.60 ± 1.05 [A,a] | 8.80 ± 1.31 [A,a] |
| | 14 | 9.50 ± 0.44 [A,a] | 8.90 ± 1.39 [A,a] | 8.70 ± 1.67 [A,a] | 9.10 ± 0.87 [A,a] | 9.00 ± 2.04 [A,a] |
| | 21 | 8.90 ± 1.09 [A,a] | 9.50 ± 0.82 [A,a] | 8.30 ± 1.50 [A,a] | 8.20 ± 1.51 [A,a] | 7.80 ± 1.95 [A,a] |
| | 28 | 9.20 ± 0.64 [A,a] | 8.30 ± 1.89 [A,a] | 8.50 ± 1.23 [A,a] | 8.90 ± 1.01 [A,a] | 8.80 ± 1.19 [A,a] |
| total | 7 | 18.70 ± 0.45 [A,a] | 19.20 ± 0.23 [A,a] | 18.2 ± 0.33 [A,a] | 18.40 ± 0.29 [A,a] | 18.40 ± 0.37 [A,a] |
| | 14 | 19.30 ± 0.18 [A,a] | 17.70 ± 0.60 [B,b] | 18.50 ± 0.39 [A,a] | 18.80 ± 0.39 [A,a] | 18.90 ± 0.42 [A,a] |
| | 21 | 18.70 ± 0.27 [A,a] | 19.20 ± 0.29 [A,a] | 17.10 ± 0.58 [B,a] | 17.10 ± 0.52 [C,b] | 16.40 ± 0.66 [D,b] |
| | 28 | 18.70 ± 0.26 [A,a] | 17.60 ± 0.48 [A,c] | 17.10 ± 0.48 [B,a] | 18.50 ± 0.30 [A,a] | 18.2 ± 0.31 [A,a] |

C—control cheese from brine with 100% NaCl; CC1—cheese from brine with 75% NaCl and 25% Ca-citrate; CC2—cheese from brine with 50% NaCl and 50% Ca-citrate; CL1—cheese from brine with 75% NaCl and 25% Ca-lactate; CL2—cheese from brine with 50% NaCl and 50% Ca-lactate [A–D] The same superscript capital letters within a row denote no significant differences ($p > 0.05$) between the values obtained for the different trials regarding the control sample according to Tukey's ANOVA. [a–c] The same superscript lowercase letters within a column denote no significant differences ($p > 0.05$) between values obtained for different days of storage according to Tukey's ANOVA.

### 3.4. PCA Analysis

The relationship between process variables and the analyzed physical, textural, and sensory properties of the brine and cheese samples was estimated using the PCA analysis. According to Silva Matinis et al., [37]. PCA is an approach that is used to identify the most important factors causing changes in industrial foods. The loading plots for the PCA analysis of the brine properties are given at Figure 3a. The sodium chloride concentration, Ca-citrate concentration, Ca-lactate concentration, and the day of storage were selected as supplementary variables, and the pH, conductivity, TDS, and color coordinates (L*, a*, b*) were used as the variables for the analysis. It can be noticed that first two PCs contribute to 75.23% of the data variability (PC 1 = 55.74% and PC 2 = 19.49%). The results show that the conductivity and TDS of brine are positively correlated with the NaCl concentration. It can also be noticed that the pH and b* coordinate of the brine samples color are positively correlated with the day of storage, while the a* coordinate of brine samples color is positively correlated with the calcium lactate concentration. Furthermore, the conductivity and TDs of the brine samples are negatively correlated with the a* coordinate of the brine samples' color. In the case of cheese samples (Figure 3b), the first two PCs contribute to 81.57% of the data variability (PC 1 = 69.43% and PC 2 = 12.14%). The brine pH, brine conductivity, brine TDS, day of storage, sodium chloride concentration, Ca-citrate concentration, and Ca-lactate concentration were selected as supplementary variables, and the pH, conductivity, TDS, and color coordinates (L*, a*, b*), cheese textural properties (hardness, adhesive force, adhesiveness, cohesiveness, gumminess, postponed elasticity, chewiness, resistance, breakage, and fibrousness) and cheese sensory properties (appearance, color, consistency, cut, odor, taste and total) were used as the variables for the analysis. According to the PCA loading plot, the cheese pH was positively correlated with the brine pH, day of storage, Ca-lactate concentration, and Ca-citrate concentration. The breakage, gumminess, hardness, chewiness, cut, titratable acidity, and postponed elasticity were positively correlated with the brine conductivity, NaCl concentration, and brine TDS, while the color, taste, appearance, consistency, odor, adhesive forces, and fibrousness

were reciprocally positively correlated. The results confirm the efficient usage of the PCA analysis for the simulation analysis of the relationships between multiple properties of cheeses samples, which agrees with data in the literature [38].

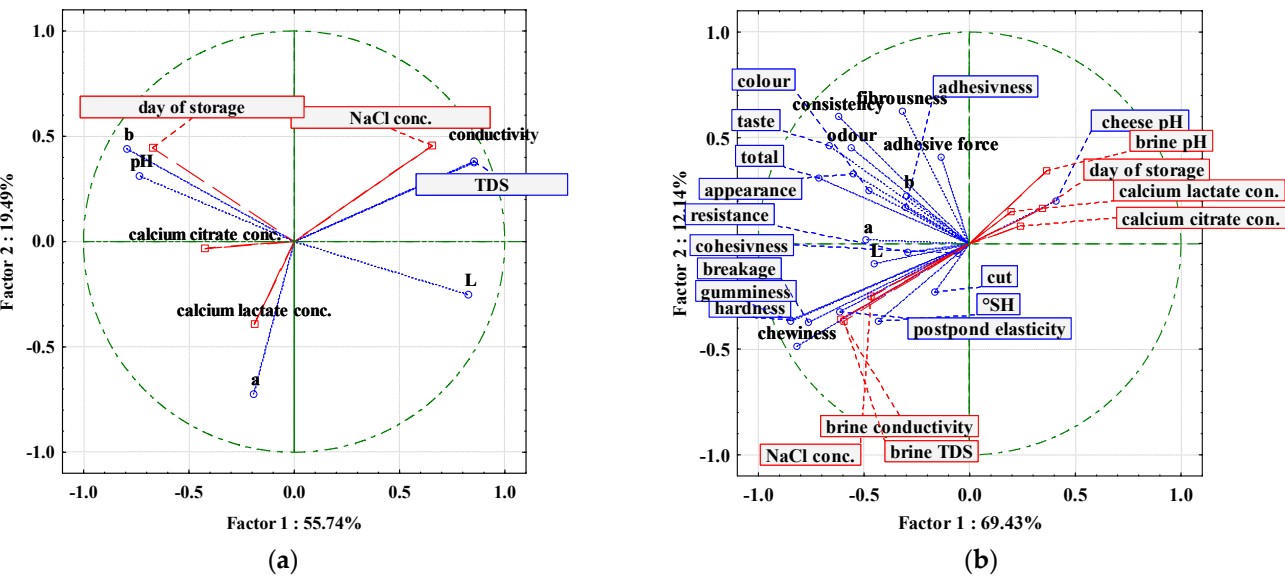

**Figure 3.** Principal component analysis (PCA) loading plots of (**a**) brine samples and (**b**) cheese samples. Blue lines present variables for analysis, while red lines present supplementary variables.

### 3.5. Artificial Neural Network Modeling

Artificial Neural Network (ANN) modeling was used to predict the brine, physical and textural properties of cheese and to conduct a sensory evaluation of the produced cheeses over the 28-day storage period. ANNs are an effective tool for tasks like prediction because of their strong learning capabilities, which enable information extraction for complicated data sets [39]. Four different ANN models were developed (i) for the prediction of brine properties (pH, conductivity, TDS and color coordinates) based on the NaCl concentration, calcium citrate concentration, calcium lactate concentration, and day of storage; (ii) for the prediction of the physical properties of cheese (pH, °SH, L and color coordinates) based on the NaCl concentration, calcium citrate concentration, calcium lactate concentration, day of storage, brine pH, and brine conductivity; (iii) for the prediction of the textural properties of cheese (hardness, gumminess, chewiness, and breakage) based on the NaCl concentration, calcium citrate concentration, calcium lactate concentration, day of storage, brine pH, and brine conductivity; and (iv) for the total sensory evaluation based on the NaCl concentration, calcium citrate concentration, calcium lactate concentration, day of storage, brine pH, and brine conductivity. The architectures of ANNs selected for the description of specific properties of brine and white brined cheese are given in Table 6.

**Table 6.** Architecture of the ANNs selected for the description of the specific properties of brine and white brined cheese.

| Sample | Network Structure | Hidden Activation Function | Output Activation Function | Training Perf. Training Error | Test Perf. Test Error | Validation Perf. Validation Error |
|---|---|---|---|---|---|---|
| Brine | MLP 4-8-6 | Exponential | Logistic | 0.9746 0.0098 | 0.9489 0.0221 | 0.9548 0.0523 |
| | | | **Output variable** | | | |
| | | | pH | 0.9558 | 0.95133 | 0.8186 |
| | | | S | 0.9963 | 0.9958 | 0.9864 |
| | | - | TDS | 0.9981 | 0.9968 | 0.987 |
| | | | L | 0.9871 | 0.9845 | 0.9805 |
| | | | a | 0.9660 | 0.9348 | 0.9336 |
| | | | b | 0.9643 | 0.9629 | 0.8702 |
| Cheese | MLP 7-7-5 | Exponential | Tanh | 0.8966 0.0294 | 0.8922 0.0317 | 0.8691 0.0558 |
| | | | **Output variable** | | | |
| | | | pH | 0.9085 | 0.9012 | 0.8089 |
| | | | °SH | 0.9346 | 0.9331 | 0.8286 |
| | | - | L | 0.9086 | 0.8744 | 0.7433 |
| | | | a | 0.9005 | 0.8608 | 0.8396 |
| | | | b | 0.8771 | 0.7418 | 0.7289 |
| | MLP 7-6-4 | Tanh | Identity | 0.9622 0.0051 | 0.8378 0.0354 | 0.8358 0.0766 |
| | | | **Output variable** | | | |
| | | | Hardness | 0.9696 | 0.8622 | 0.8406 |
| | | - | Gumminess | 0.9525 | 0.8325 | 0.7754 |
| | | | Chewiness | 0.9754 | 0.9655 | 0.8553 |
| | | | breakage | 0.9514 | 0.9483 | 0.8149 |
| | MLP 7-6-1 (total sensory) | Tanh | Identity | 0.9354 0.0035 | 0.9357 0.0045 | 0.8213 0.0135 |

To predict the physical properties of brine, the MLP 4-8-6 network with the hidden activation function Exponential and the Logistic function as the output activation function was selected. The learning, testing, and validation $R^2$ values for this model were 0.9746, 0.9489, and 0.9548, and the corresponding RMSE values were 0.0098, 0.0221, and 0.0523. The selected ANN model describes all analyzed model outputs with high precision. The best agreement between the experimental data and ANN-model-predicted data for the prediction of brine physical properties was obtained for the TDS ($R^2_{validation}$ = 0.9870) (Figure 4b), followed by S ($R^2_{validation}$ = 0.9864) (Figure 4c) and L* ($R^2_{validation}$ = 0.9805) (Figure 4d). On the other hand, the lowest agreement between the experimental data and the ANN-model-predicted data was obtained for the pH ($R^2_{validation}$ = 0.8186) (Figure 4a). To predict the physical properties of cheese, the MLP 7-7-5 network with the hidden activation function Exponential and the Tanh as the output activation function was selected. The learning, testing, and validation $R^2$ values for MLP 7-7-5 were 0.8966, 0.8922, and 0.8691, respectively. The results show that MLP 7-7-5 most accurately predicted the a* coordinate of color ($R^2_{validation}$ = 0.8696) (Figure 5d) and titratable acidity ($R^2_{validation}$ = 0.8686) (Figure 5b), while the lowest agreement between experimental data and model-predicted

data was obtained for the b* coordinate of color ($R^2_{validation}$ = 0.7289) (Figure 5e). The MLP 7-6-4 network with the hidden activation function Tanh and the Identitiy function as the output activation function was selected for the prediction of the textural properties of cheese. The results showed that the developed ANN model efficiently described the cheese chewiness ($R^2_{validation}$ = 0.8553) (Figure 5h), cheese hardness ($R^2_{validation}$ = 0.8406) (Figure 5f), and cheese breakage ($R^2_{validation}$ = 0.8149) (Figure 5i). Due to the importance of the sensory evaluation during the development of new food products, the applicability of ANN modeling for the prediction of the total sensory score based on the NaCl concentration, calcium citrate concentration, calcium lactate concentration, day of storage, brine pH, and brine conductivity was assessed. MLP 7-6-1 was selected as the optimal network for the prediction of the total sensory score. The MLP 7-6-1 network had Tanh as the activation function and the Identity function as the output activation. The learning, testing, and validation $R^2$ values for this model were 0.9354, 0.8357, and 0.8213, and the corresponding RMSE values were 0.0035, 0.0045, and 0.0135 (Figure 5j). The presented results show the significant potential of using ANN modeling for the prediction of cheese properties.

As described by several authors [40–43] ANN modeling is a popular statistical tool in food sciences because of its benefits, which include the capacity to learn from examples, fault tolerance, real-time operation, and the ability to forecast nonlinear data. There are some examples of ANN modeling usage for the prediction of cheeses' specific properties. Goyal and Goyal [44] developed Artificial Neural Network models to forecast the length that processed cheese will last when it is stored at 7–8 °C. For the development of the models, the following factors were used as inputs: body and texture, fragrance and flavor, moisture, and free fatty acids. The output parameter was the sensory score. We compared the created Cascade single layer ANN models to one another. The ANN models were trained using Bayesian regularization. The shelf life of processed cheese was effectively predicted using cascade ANN models. Vasques et al. [45] used ANN models for the evaluation of the Swiss-type cheese texture during the ripening process based on hyperspectral images, and their results showed that the developed ANN model could predict hardness during maturity with high accuracy ($R^2$ = 0.97). Furthermore, Stangierski et al. [46] compared the efficiency of multiple regression models and ANN models as prediction tools for changes in the overall quality during the storage of Gouda cheese based on five factors selected by the PCA analysis. The cheeses' physical properties during storage were variables in the PCA analysis, and the authors showed that the ANN model with $R^2$ 0.99 could precisely predict quality deterioration during storage, which is significant for assessing the risk or the safety and quality of food. The feedforward–backpropagation ANN model was also efficiently applied to predict the sensory scores in Indian cheeses (paneer) with different moisture levels (40–50%) [47]. Moreover, Morita et al. [48] used the ANN model to demonstrate the nonlinear relationships among the raw gas chromatography/olfactometry data of the Cheddar cheese samples and the palatability scores for the same cheese samples.

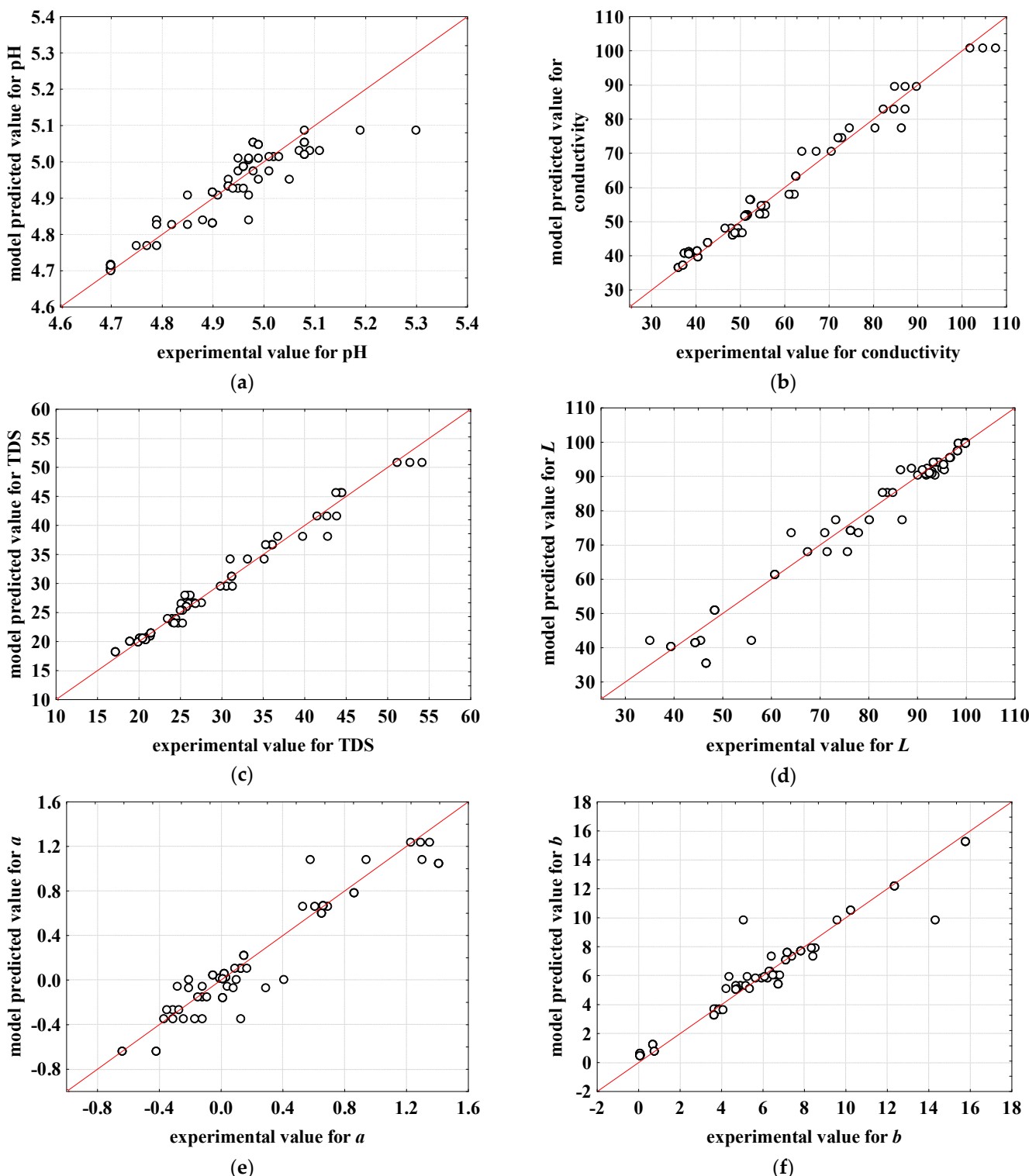

**Figure 4.** Comparison between experimental data and ANN-model-predicted data for some brine. properties. (**a**) pH value; (**b**) conductivity; (**c**) total dissolved solids; (**d**) color—L*; (**e**) color—a*; (**f**) color—b*.

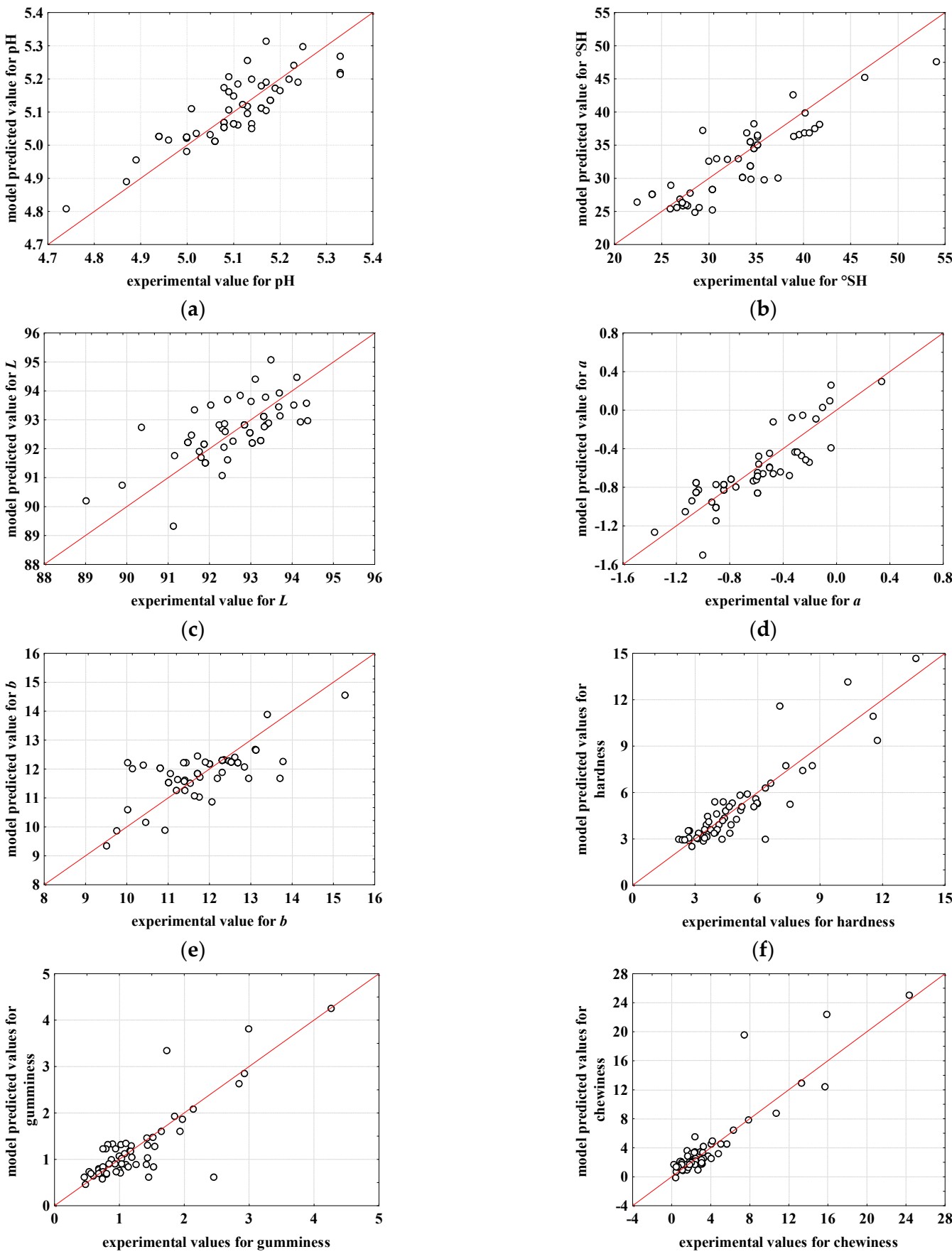

**Figure 5.** *Cont.*

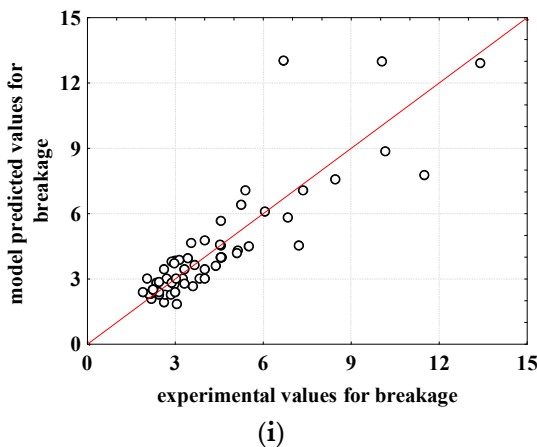
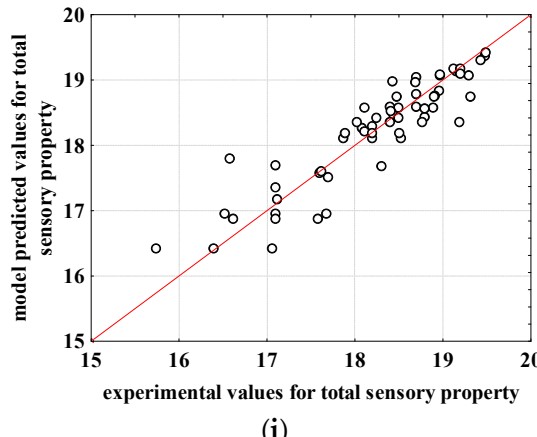

**Figure 5.** Comparison between experimental data and ANN-model-predicted data for some cheese properties. (**a**) pH value; (**b**) titratable acidity (°SH); (**c**) color-L*; (**d**) color-a*; (**e**) color-b*; (**f**) texture—hardness; (**g**) texture—gumminess; (**h**) texture—chewiness; (**i**) texture—breakage; (**j**) texture—total sensory score.

## 4. Conclusions

The replacement salts Ca-lactate and Ca-citrate have excellent potential for use in sodium-reduced white brined cheese production. The results showed that the replacement of 25% NaCl with substitutional salts did not influence the physical–chemical, textural, sensory, and color properties of cheese. The best sensory score was gained with the substitution of 25% NaCl with Ca-citrate. When a 50% salt substitution was used in brine, the produced cheese had a softer texture and was more breakable, and panellists described the cheese as having a gritty mouthfeel. ANN prediction models can easily predict brine properties and the physical and textural properties of cheese and can be used to conduct a total sensory evaluation based on the NaCl concentration, calcium citrate concentration, calcium lactate concentration, day of storage, brine pH, and brine conductivity.

**Author Contributions:** Conceptualization, K.L.J., I.B.J. and A.J.T.; Methodology, K.L.J., I.B.J. and A.J.T.; Software, A.J.T.; Validation, I.B.J., K.L.J. and A.J.T.; Formal Analysis, K.L.J., I.B.J. and N.M.R.; Investigation, K.L.J., I.B.J. and N.M.R.; Resources, R.B.; Data Curation, K.L.J., I.B.J. and A.J.T.; Writing—Original Draft Preparation, K.L.J., I.B.J. and A.J.T.; Writing—Review & Editing, K.L.J., I.B.J. and N.M.R.; Visualization, K.L.J.; Supervision, K.L.J.; Project Administration, K.L.J.; Funding Acquisition, I.B.J. All authors have read and agreed to the published version of the manuscript.

**Funding:** This research was funded by the University of Zagreb, Grant name: Cheese production with reduced salt amount.

**Institutional Review Board Statement:** Not applicable.

**Informed Consent Statement:** Not applicable.

**Data Availability Statement:** Not applicable.

**Conflicts of Interest:** The authors declare no conflict of interest.

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
