# Peer review of "Reduced Sodium in White Brined Cheese Production: Artificial Neural Network Modeling for the Prediction of Specific Properties of Brine and Cheese during Storage"

_fermentation, doi:10.3390/fermentation9090783_

Round 1

Reviewer 1 Report

The work is very interesting and it is justified from the technological and health aspects. The authors clearly and very precisely described the influence of the partial substitution of sodium chloride in feta cheese with other salts on the physical-chemical, textural and sensory characteristics. This paper  is statistically very precise discussed, but I suggest the authors to explain the differences and changes in textural characteristics between the samples in a little more detail and connect them with other physicochemical parameters such as pH value and acidity (page 10, line 365).

After a minor correction,  the paper could be published . 

Author Response

Respected Reviewer,

I sincerely appreciate your valuable comments and insightful suggestions. Your input has greatly contributed to enhancing the quality of the Manuscript.

I have taken into account the differences in textural properties and their correlation with cheese characteristics, and I have incorporated these changes into the manuscript. Please refer to page 12, lines 404 - 419.

I have attached the revised manuscript for your perusal.

Thank you once again for your time and guidance.

Best regards

Reviewer 2 Report

Dear authors,

Your submitted paper is interesting in general, with soundness. The paper is well designed and the results are presented appropriately. 

However, before acceptance I suggest to solve some serious issues about the labelling of the testedcheese type. As you probably know, Feta is a PDO cheese and its production is presented by legislation rules. You should change the name of your cheese tested from Fera type cheese into White brined cheese. Your cheese is not Fera type, because you have used cows milk, which is a type of milk not used in Feta cheese production. For example Těleme or Telemea (and other cheese types) are produced from cows milk. Make this change in the whole manuscript,  from title, abstrakt, methods, duscussion to conclusion.

Minor corrections 

Author Response

Respected Reviewer,

Thank you very much for your comment and valuable suggestion for our manuscript.

Your suggestion has proven to be very valuable, and we have made the change in the manuscript from 'Feta-type' to 'White brined cheese'.

Please find the revised manuscript attached.

We truly appreciate your time and guidance.

Best regards!

Round 2

Reviewer 2 Report

Dear authors,

Thank you for the corrections made.

Beer regards